# Breaking barriers: Validation of a Spanish oral health knowledge tool to enhance patient-provider communication

**Silvia Spivakovsky**[1]*, **Joyce Figueroa**[2], **Ryan Richard Ruff**[3]

**1** Department of Oral & Maxillofacial Pathology, Radiology & Medicine, New York University College of Dentistry, New York City, United States of America, **2** New York University College of Dentistry, New York City, United States of America, **3** Division of Community Oral Health, University of Pennsylvania School of Dental Medicine, Philadelphia, United States of America

☯ These authors contributed equally to this work.

* s.spivakovsky@nyu.edu

## Abstract

### Objectives

This study aimed to develop and validate the Knowledge Related to Oral Health Literacy Spanish (KROHL- S) instrument to assess oral health knowledge among Spanish-speaking adults in the United States, a population facing significant oral health disparities.

### Design

A cross-sectional study was conducted at NYU College of Dentistry. A convenience sample of 175 self-identified Spanish-speaking adults (70% female, mean age 49. 79 years) completed the orally administered KROHL- S questionnaire. Participants, mainly born outside the US (91. 9%), also completed the Comprehensive Measure of Oral Health Knowledge (CMOHK) and a single-item literacy screening tool in Spanish (SILS). Psychometric properties of the KROHL- S, including internal consistency (Cronbach's alpha), discriminant validity (correlation with CMOHK), and known-group validity (comparison across education levels), were evaluated. Confirmatory factor analysis was used to test the original factor structure.

### Results

The mean KROHL- S score was 8.34 (SD = 5.82), indicating a low level of oral health knowledge in the sample. Internal consistency for the overall KROHL- S was good (Cronbach's alpha = 0.75), and interrater agreement was high. A moderate positive correlation was found between KROHL- S and CMOHK scores (r = 0.49, p < .0001). Participants with higher education levels showed significantly greater oral health

**Data availability statement:** The data is available through the following link: https://doi.org/10.58153/frv59-hzf81.

**Funding:** The author(s) received no specific funding for this work.

**Competing interests:** The authors have declared that no competing interests exist.

knowledge on the KROHL- S. Confirmatory factor analysis suggested an average fit to the data (RMSEA = 0.064, CFI = 0.86, TLI = 0.83).

## Conclusion

The KROHL- S could be used to assess oral health knowledge among Spanish-speaking adults and incorporates cultural and linguistic aspects, making it suitable for a wider range of individuals. KROHL-S offers a valuable tool for health-care providers by not only helping identify individuals' knowledge gaps to guide customized educational interventions but also helping enhance patient-provider communications.

## Introduction

The Hispanic population is the largest minority group in the US. Although clustered under a single ethnic group, individuals who self-identify as Hispanic include people from at least 19 Latin American countries, with nearly a third being foreign-born [1]. Despite this demographic significance, Hispanic communities experience disproportionately high rates of dental disease, including caries, periodontal disease, and tooth loss [2]. These disparities stem from multiple social determinants, including socio-economic status, language barriers, cultural differences, limited access to healthcare services, and varying levels of health literacy [2–4].

Oral health literacy is defined as one's ability "to obtain, process, and understand basic oral health information and services needed to make appropriate health decisions" [5]. Research consistently demonstrates a strong correlation between limited health literacy and poor oral health outcomes [2,6]. Spanish-speaking individuals with low health literacy are significantly more likely to miss routine dental check-ups, report difficulty understanding dental treatment instructions, and are less likely to engage in preventive oral health behaviors [2]. Different levels of educational attainment and limited access to oral health care in many areas of Latin America are further contributors [2,7].

To better understand the role of health literacy and its contributing factors to the oral health of our Spanish-speaking patients, our team developed a working framework that led to the creation of the KROHL-S model. The model was presented elsewhere [8]. In brief, the KROHL-S model was designed to help identify clinically relevant factors and relationships that are practical at the point of care, specifically those that can be improved or modified through patient-provider interactions. Cultural factors, self-efficacy, fear, and health literacy all play an important role in the development of effective patient-centered communications, which demonstrated to have a direct impact on oral health outcomes [9].

After the development of the conceptual model, we proceed to identify those tools that will be used during implementation. Our initial review of existing tools [8] to evaluate oral health literacy demonstrated that the most widely studied tools rely on word pronunciation (REALD-30, REALD-66, REALM-D, REALMD-20, and TS-REALD).

Other instruments use a combination of word recognition and comprehension (OHLIP, OHLA-E/OHLA-S, and MOHLR-K), a combination of comprehension and numeracy (TOHFLID, OHLI, and SOHLS), or decision-making and communication levels are part of the skills evaluated by HelD29/HelD-14 and OHL-AQ, and one tool evaluated conceptual knowledge (CMOHK).

The decision to evaluate oral health knowledge rather than oral health literacy was supported by the body of evidence linking health literacy, health knowledge, understanding of instructions, and correct disease-specific beliefs in medicine [10–11] as well as in oral health [12–13]. In addition, changes in health knowledge are a parameter widely used to assess public health interventions [14–16].

Moreover, there is no gold standard to measure oral health literacy [17–19], there is evidence questioning the quality of translations and cross-cultural adaptation of oral health literacy tools for non-English speakers [19], and the uncertain reliability of word recognition tests in general [20] and specifically in a language with phoneme-grapheme concordance [18,19,20} With health literacy considered content and context specific [21], the assessment of oral health knowledge was identified as the most useful aspect of personal oral health literacy that could be evaluated at the point of care, moving away from "labeling" a patient's literacy level and instead identifying concrete patient educational needs.

The choice of a tool with non-reading requirements was intentionally selected to address some of the issues reported, leading to poor participation of Hispanics in research [22] and the scope of the population for whom it may be applicable [8]. This choice also reflects the shifts in information acquisition, as more individuals rely on digital sources, social media, news websites, and podcasts. [23]. As such, text-based literacy may not fully capture how individuals obtain information.

The study aimed to develop and test the psychometric properties of the Knowledge Related to Oral Health Literacy scale for Spanish speakers. (KROHL S). In the clinical setting, this tool can contribute to patient-centered care by identifying knowledge gaps that can be translated into personalized and linguistically appropriate interventions to improve communication and oral health.

## Materials and methods

### Research design and participants

This was a cross-sectional study approved by the New York University School of Medicine Institutional Review Board (FY2019-2790). Data were collected from Spanish speakers at the New York University College of Dentistry. Potential participants were recruited from clinical waiting areas within the college. Eligibility criteria required participants to be at least 18 years old and to be Spanish speakers. Individuals who met the inclusion requirements were invited to take part in the study, and the objectives and procedures were explained in detail. Following Institutional Review Board (IRB) approval, oral consent was obtained from everyone who agreed to participate after ensuring their understanding of the study. The consent process was witnessed and documented, and a written copy was provided to the participant. Oral consent was chosen to accommodate potential literacy challenges and to help preserve anonymity.

From 06/15 2019 to 03/10/2020, of the 185 individuals approached,175 participants were enrolled and completed the questionnaire. Most participants were patients, although some were parents accompanying patients. The questionnaire was administered through face-to-face interviews in a private setting by trained interviewers who recorded the participants' responses. A printed copy of the survey was offered as well to view the questions. Participants were instructed that it was perfectly acceptable to respond with "I don't know" if they were uncertain. Each interview lasted approximately 20 minutes. All data was collected using Qualtrics software.

### The KROHL-S questionnaire

As in the English version, the KROHL-S questionnaire consisted of four questions each for five separate conditions: caries, periodontal disease, oral cancer, tooth loss, and malocclusion. Each question is related to a specific content

domain: identification and symptoms, causes, prevention, and treatment. Each question was open-ended and followed a standard format:

1. "What do you see and feel while having _______ (condition options: caries, periodontal disease, oral cancer, tooth loss or malocclusion)?"

    a. "¿Qué es lo que ve o siente cuando tiene ------ (caries, enfermedad de las encías, cáncer oral, perdida de dientes o maloclusión)?"

2. "What causes _______ (condition)?"

    a. "¿ Qué causa -----------(nombre de la condicion)?"

3. "How do you prevent _______ (condition)?"

    a. "¿Qué se puede hacer para prevenir ---------- (nombre de la condicion)?"

4. "How do you treat ______ (condition)?"

    a. "¿ Cual es o son los tratamientos de -------(nombre de la condicion)?"

The questionnaire was developed in Spanish by a native Spanish speaker. Five non-team native Spanish speakers from different countries then reviewed the questionnaire for readability and provided recommendations on included terminology. In addition to domain-specific items, demographic information (e.g., age, sex, place of birth, years living within the United States, level of education, insurance status, and oral health status) was also collected.

Following data collection, trained evaluators who were native Spanish speakers evaluated each response for scoring. The scoring system was built following the same approach as with the English version of the KROHL [24]. Team members agreed on the content areas that should be used as proof of optimal knowledge for each question, and from that starting point, a set of parameters was developed and tested until reaching consensus. Based on the pre-established set of parameters, the most commonly used words and their synonyms provided by the participants when answering to a particular question were used to develop the detailed scoring criteria. For example, when assessing caries prevention, three content areas were assessed: oral self-care practices, dietary considerations, and professional care. The highest score was given to an answer that include brushing teeth after a meal ("cepillarse despues de comer"), don't eat sweets ("no comer dulces") and go to the dentist ("ir seguido al dentista").

Responses to each question were then graded using a 4-point scale: good (2 points), fair (1 point), minimal (0.5 point), or no points (0) if the response was 'I don't know' or incorrect. The final KROHL-S score was the sum of the 4 individual questions for each of the 5 conditions. The question on see/feel for tooth loss was excluded from the final calculation because, as with the English version, patients only rephrased the concept of tooth loss.

### Additional measures

Data were also collected using the Comprehensive Measure of Oral Health Knowledge (CMOHK) scale and the Single-item literacy screening tool in Spanish (SILS). The CMOHK [25] consists of 23 multiple-choice questions assessing general knowledge of oral health, including the prevention and management of dental caries, periodontal disease, and oral cancer. The CMOHK, which is also available in Spanish, has shown strong psychometric reliability across various settings, including Spanish speakers [26]. Scores are calculated by totaling the number of correct responses. It categorized individuals as having poor (0–11), fair (12–14), or good levels (15–23) of oral health knowledge. A modified scoring system is also available with 2 categories: low (0–14) or good (15–23) [26]. The single-item literacy screening (SILS) tool in Spanish uses a 5-point Likert scale to answer: "How confident are you filling out medical forms by yourself? (¿Con qué frecuencia puede usted completar formularios de salud sin recibir ayuda?), with 1 indicating never (nunca) and 5 indicating

always (siempre). SILS has effectively identified individuals with inadequate health literacy among Spanish speakers in various settings [27,28].

## Statistical analysis

Descriptive statistics were computed for the full KROHL-S scale and each individual subscale, and histograms were produced. Internal consistency was evaluated using Cronbach's alpha, and discriminant validity was assessed using Pearson's correlation comparing KROHL-S scores to CMOK scores. Interrater agreement was calculated using the Intraclass Correlation Coefficient. Known-group validity was explored by comparing scores by level of education using one-way analysis of variance with post-hoc tests using the Tukey multiple comparisons of means method. The preexisting factor structure of the English-language version of the KROHL was then tested in the new sample using confirmatory factor analysis.

**Sample size rationale.** Conventional sample size requirements for factor analyses were followed [29]. There were no instances of missing data and no convergence issues in the analysis.

## Results

Of the 175 enrolled participants, 70.85% were female, with an average age of 49.79 (ranging between 18 and 85 years). The majority (92%) were born outside the US, representing 13 Latin American countries, and had lived in the US for an average of 22.72 years (range from 1 to 60 years). Spanish was the primary language for 97.17% of participants. Approximately 37.14% of participants reported not having insurance to cover for their dental treatment. Regarding dental visits, 70.85% had visited a dentist within the past year, 18.28% had done so between 1 and 2 years, and 10.85% had not visited a dentist for more than 2 years. Notably, 2 of the participants were attending their first-ever dental visit (Table 1).

A histogram of scores by subscale and overall is shown in Fig 1. For DC sub-scores, 11.4% of respondents had a 0 score. Similarly, 34.9% of respondents scored 0 on GD, 62.9% scored 0 on OC, 26.9% scored 0 on TL, 49.1% scored 0 on MO. Overall, 9.7% of respondents scored 0 on the total KROHL instrument.

Self-reported oral health status was rated as "good" or "very good" by 70.27% of the participants. For educational attainment, 36.71% had completed college or postgraduate education, 40.67% had a high school diploma, 15.25% had completed elementary school, and 7.21% had not completed elementary school.

The average KROHL-S score across all participants was 8.34 (SD = 5.82), with the greatest level of knowledge found in dental caries and the lowest in oral cancer. The average score for the CMOHK was 12.05, ranging from 2 to 21. Around 60% of participants received a score of 14 or less, which is considered "low level" for oral health knowledge.

Internal consistency (Table 2) was good for the overall scale (0.85) and oral cancer (0.74) and malocclusion (0.72) subscales, and acceptable for dental caries (0.60) and periodontal disease (0.68). However, Cronbach's alpha for tooth loss was 0.53. Interrater reliability was high (ICC = 0.99, 95% CI = 0.99, 0.99, p < .0001).

The correlation between the KROHL-S and CMOHK instruments was 0.49 (p < .0001), suggesting a moderate degree of similarity between the two scales. Generally, respondents who received a college or graduate education had higher overall and disease-specific knowledge compared to those with less than elementary, elementary level, or high school level education (Table 3).

Confirmatory factor analysis resulted in a root mean square error of approximation of 0.064, and a Comparative Fit Index (CFI) and Tucker-Lewis Index (TLI) of 0.86 and 0.83, respectively. However, the RMSEA was statistically significant at 0.05 (p = 0.044). The resulting CFA plot is shown in Fig 2.

## Discussion

Oral health knowledge, as an important component of health literacy, is considered a potentially modifiable factor influencing social determinants of health and, therefore, could be a pathway to improve health disparities [21]. The purpose of the study was to develop an instrument to evaluate oral health knowledge among Spanish speakers. Psychometric analysis

**Table 1. Participants characteristics.**

| Age | 18-29 | 22 | 12.57% |
|---|---|---|---|
| | 30-39 | 27 | 15.43% |
| | 40-49 | 36 | 20.57% |
| | 50-59 | 37 | 21.14% |
| | 60-69 | 32 | 18.28% |
| | 70 and older | 21 | 12.00% |
| Sex | Female | 124 | 70.85% |
| | Male | 51 | 29.14% |
| Country of birth | US | 14 | 8% |
| | Other | 161 | 92% |
| Years in the US | Average | 22.72 years | from 1 to 60 |
| Main language | Spanish | 170 | 97.17% |
| | English | 5 | 2.82% |
| Highest level of education | post College | 9 | 5.14% |
| | College | 56 | 32.00% |
| | HS | 72 | 41.14% |
| | Elementary | 27 | 15.43% |
| | Incomplete Elementary | 12 | 6.85% |
| Oral health status self-reported | Very poor | 14 | 8.00% |
| | Poor | 38 | 21.71% |
| | Good | 97 | 55.42% |
| | Very good | 26 | 14.85% |
| Dental insurance | Yes | 110 | 62.85% |
| | No | 65 | 37.14% |
| Last dental visit | <1 year | 124 | 70.85% |
| | Between 1 and 2 years | 32 | 18.28% |
| | >2 but <than 4 years | 11 | 6.28% |
| | ≥4 years | 8 | 4.57% |
| confident filling forms w/o help | Always | 29 | 16.66% |
| | Almost always | 27 | 15.51% |
| | Half of the time | 22 | 12.64% |
| | Almost never | 40 | 22.98% |
| | Never | 56 | 32.18% |

\* Not all totals equal 175 due to missing data.

showed that the KROHL-S had a high internal consistency nearly identical to that of the English-language version, that adults with higher levels of education demonstrated greater knowledge of oral disease, and that there was a significant moderate correlation with CMOHK scores. However, the fit of the overall scale structure was average.

We found that a diverse, cross-sectional sample of Spanish-speaking adults had a low level of oral health knowledge, with average KROHL-S scores being <10 compared to the maximum score of 38. Over 60% of respondents also scored low on the CMOHK. Compared to the sample used in validating the English-language version of the KROHL [24], native Spanish speakers scored an average of 5 points lower on the overall scale. These results are similar to other studies assessing oral health knowledge in Hispanics [26], in which average scores on the Comprehensive Measure of Oral Health Knowledge scale were low. Additionally, a prior study validating the Short Test of Functional Health Literacy in

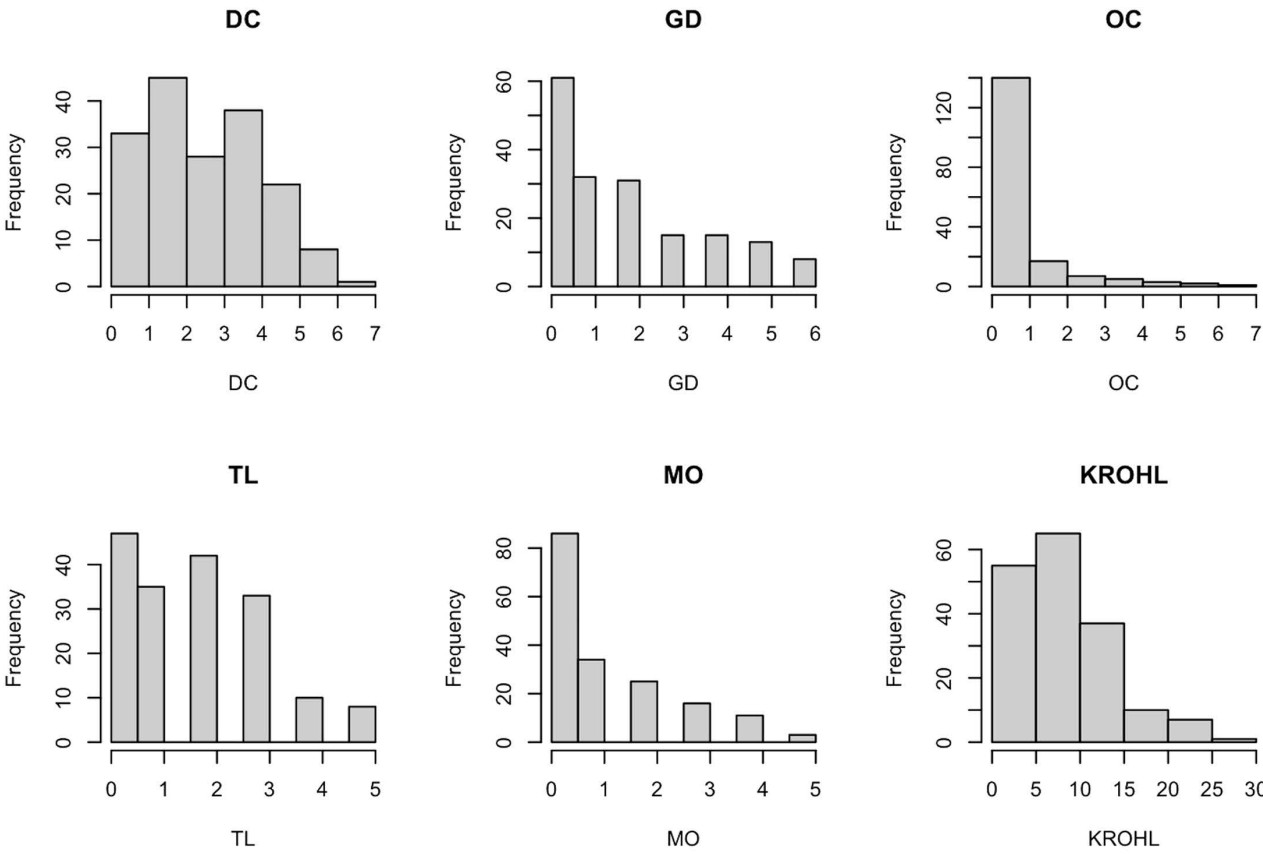

**Fig 1. Histogram of scores.**

**Table 2. Descriptive Statistics for KROHL-S scale and subscales.**

|  | Mean | SD | Min | Max | Alpha |
|---|---|---|---|---|---|
| Full | 8.34 | 5.82 | 0 | 26 | 0.85 |
| Dental caries | 2.88 | 1.68 | 0 | 7 | 0.6 |
| Periodontal disease | 1.78 | 1.84 | 0 | 6 | 0.68 |
| Oral cancer | 0.79 | 1.37 | 0 | 7 | 0.74 |
| Tooth loss | 1.7 | 1.42 | 0 | 5 | 0.53 |
| Malocclusion | 1.09 | 1.36 | 0 | 5 | 0.72 |

Adults (Short-TOFHLA) found that 58% of Spanish speakers had inadequate scores, compared to only 2.8% of English speakers [28].

The psychometric properties of the KROHL-S are similar to other measures of oral health literacy for Spanish-speaking populations. The 23-item CMOHK was found to have an overall Cronbach's alpha of 0.72, and CMOHK scores were significantly related to educational level. Additionally, there was a significant correlation between CMOHK and the Rapid Estimate of Adult Literacy in Medicine (REALM), the latter of which being similarly correlated with the SHORT-TOFHLA [25]. Finally, validation of the Oral Health Literacy Assessment in Spanish (OHLA-S) had high levels of internal consistency

**Table 3. KROHL-S scale and subscale by level of education.**

**KROHL scale and subscale by level of education***

*significant post-hoc comparisons only

| | | Mean Diff | 95% Lower | 95% Upper | p-value |
|---|---|---|---|---|---|
| Dental caries | 4 vs 1 | 1.76 | 0.29 | 3.22 | 0.01 |
| | 5 vs 1 | 2.23 | 0.24 | 4.23 | 0.02 |
| | 4 vs 2 | 1.04 | 0.004 | 2.09 | 0.048 |
| Periodontal disease | 4 vs 2 | 1.34 | 0.19 | 2.49 | 0.01 |
| Oral cancer | 4 vs 2 | 1.04 | 0.2 | 1.88 | 0.01 |
| | 5 vs 2 | 1.52 | 0.14 | 2.9 | 0.02 |
| | 4 vs 3 | 0.73 | 0.09 | 1.37 | 0.12 |
| Malocclusion | 4 vs 1 | 1.44 | 0.24 | 2.64 | 0.01 |
| Total | 4 vs 1 | 6.57 | 1.59 | 11.56 | 0.003 |
| | 5 vs 1 | 7.56 | 0.76 | 14.35 | 0.02 |
| | 4 vs 2 | 4.83 | 1.29 | 8.37 | 0.002 |
| | 4 vs 3 | 2.72 | 0.03 | 5.42 | 0.046 |

Note: 1 = incomplete elementary, 2 = elementary education, 3 = high school education, 4 = college education, 5 = post-college education.

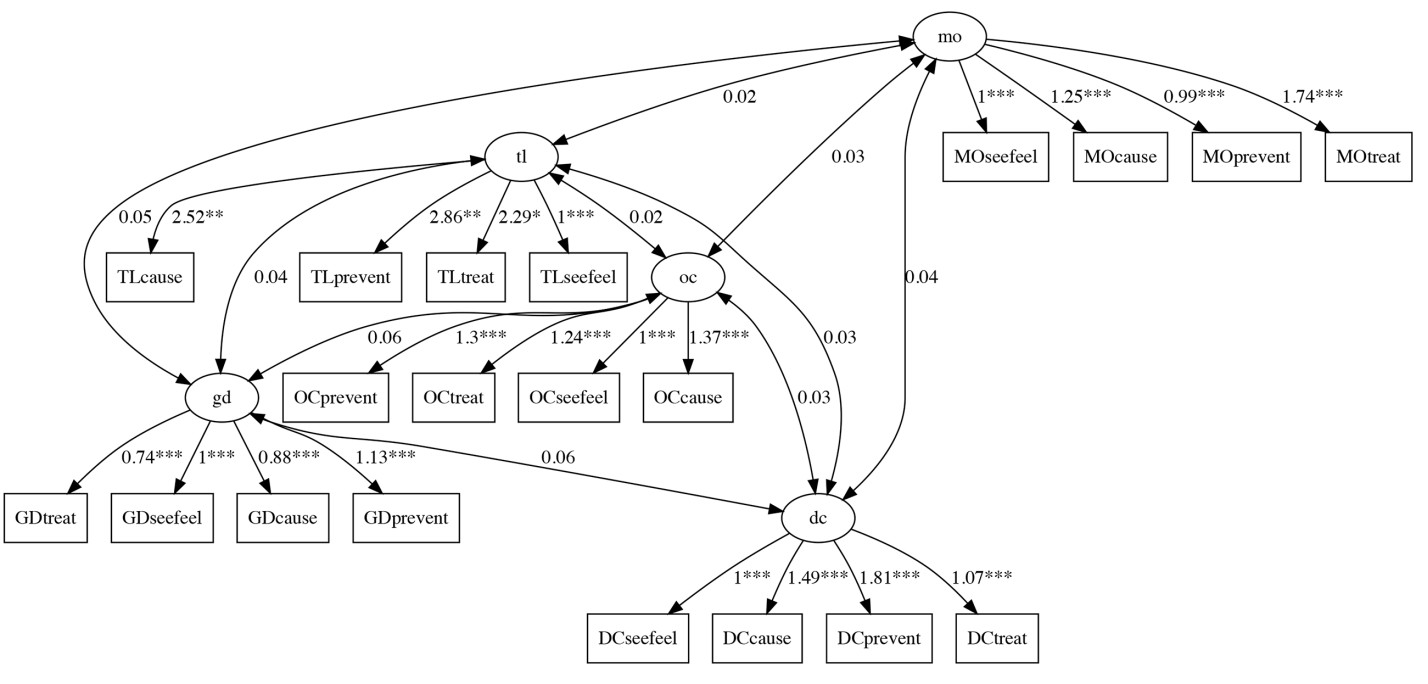

**Fig 2. CFA plot.**

(Cronbach's alpha 0.7–0.8) [30], with scores being significantly correlated with educational attainment as well as with the Short Assessment of Health Literacy for Spanish-speaking Adults (SAHLSA) [30].

KROHL-S adheres to the concept of health literacy as content and context-specific [21]. In contrast to other oral health literacy measures, KROHL-S evaluates condition-specific and domain of knowledge-specific aspects of the most common

oral conditions. Additionally, existing tools, such as the OHLA-S [31], evaluate oral health literacy among Spanish speakers based on a limited number of skills, mostly reading and comprehension. As has been shown, certain assumptions of English-language tools do not hold in the Spanish language or vice versa, such as phoneme-grapheme correspondence [31], which can limit validity in translated tools. In contrast, KROHL-S does not require reading, and pronunciation is neither proof of comprehension nor knowledge. Having no reading requirement was essential in the development of KROHL-S, as a tool that could be easily used by all kinds of individuals was desired. With the clinical setting in mind, we chose to concentrate on oral health knowledge because it is the aspect of health literacy that can be potentially modified in the clinical setting.

This study has a number of limitations. First, there are intrinsic limitations resulting from the inclusion of a single-center convenience sample. However, the diverse nature of our Institution's patient base should be acknowledged. Even within this sampling approach, we captured a broad distribution of educational backgrounds and a rich international representation spanning 13 Latin American nations. Second, evaluating patient responses to the KROHL-S requires transcription and manual evaluation of open-ended responses, relying on a scoring rubric based on expert reviewers. While this is critical to maximize accuracy in assessment, as over half of the study participants reported not being able to fill out forms without assistance, it may limit its utility in clinical practice in its actual form.

Finally, the complexity of the CFA model fit to the data may have resulted in instability of estimated fit indices given the achieved sample size, which was average considering the number of factors and items. However, many of the item loadings were large and statistically significant. As a result, we did not explore any modification indices and results should be considered as initial confirmatory evidence of the factor structure of the KROHL-S.

Artificial intelligence has the potential to overcome two major drawbacks of our current approach: capturing answers and grading. The widespread adoption of AI across all aspects of healthcare is a clear indicator of this potential. Safety, privacy, reliability, and ethical considerations are raised regarding any AI application in healthcare, and careful validation, training, and ongoing monitoring are needed to ensure its accuracy, safety, and effectiveness in supporting the delivery of care [32]. Related to our project is the rising adoption of AI applications that transcribe real-world conversations between patients and healthcare providers directly into the electronic health record [32–34]. Although widely available, they will require in-depth testing to address the raised considerations and were not part of this study.

Additionally, there is evidence of AI's accurate scoring capabilities when used in healthcare education [35,36]. At present, we are in the process of training different large language models that could be used for reliable and valid scoring.

## Conclusion

The KROHL-S can potentially provide healthcare providers with a reliable means to assess the patient educational needs (PEN) related to oral health in Spanish-speaking populations. Furthermore, KROHL-S could transcend literacy barriers through oral administration, making it particularly relevant for a diverse population, including those with limited formal education. It also could provide a new building block that can contribute to the research into the factors driving oral health disparities, although future studies with larger, more representative cohorts are needed to further validate the current findings.

## Acknowledgments

The authors would like to thank the participants for the valuable insights provided.

## Author contributions

**Conceptualization:** Silvia Spivakovsky, Ryan Richard Ruff.

**Data curation:** Silvia Spivakovsky, Joyce Figueroa, Ryan Richard Ruff.

**Formal analysis:** Silvia Spivakovsky, Ryan Richard Ruff.

**Investigation:** Joyce Figueroa.

**Methodology:** Silvia Spivakovsky, Ryan Richard Ruff.

**Project administration:** Silvia Spivakovsky, Ryan Richard Ruff.

**Validation:** Ryan Richard Ruff.

**Writing – original draft:** Silvia Spivakovsky, Joyce Figueroa, Ryan Richard Ruff.

**Writing – review & editing:** Silvia Spivakovsky, Ryan Richard Ruff.

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
