## [Decision Letter · Decision Letter 0]

19 Dec 2025

PONE-D-25-30680Breaking barriers:validation of a Spanish oral health knowledge tool to enhance patient-provider communicationPLOS One

Dear Dr. Spivakovsky,

Thank you for submitting your manuscript to PLOS ONE. After careful consideration, we feel that it has merit but does not fully meet PLOS ONE’s publication criteria as it currently stands. Therefore, we invite you to submit a revised version of the manuscript that addresses the points raised during the review process.

We look forward to receiving your revised manuscript.

Kind regards,

Hadi Ghasemi

Academic Editor

PLOS One

Journal Requirements:

2. Please include your tables as part of your main manuscript and remove the individual files. Please note that supplementary tables (should remain/ be uploaded) as separate "supporting information" files

3. Thank you for providing your underlying data as Supporting Information.

We note that the data set contains text or data that is not in English. Please note that PLOS is an English-language publisher, so we require data sets to be provided in English as well. Please upload an English-language version of your data set.

This will also allow us to determine if your data follows PLOS standards per our Data Availability policy here: https://journals.plos.org/plosone/s/data-availability

Reviewers' comments:

Reviewer's Responses to Questions

**Comments to the Author**

1. Is the manuscript technically sound, and do the data support the conclusions?

Reviewer #1: Yes

Reviewer #2: Yes

Reviewer #3: Partly

2. Has the statistical analysis been performed appropriately and rigorously? 

Reviewer #1: Yes

Reviewer #2: Yes

Reviewer #3: Yes

3. Have the authors made all data underlying the findings in their manuscript fully available?

Reviewer #1: Yes

Reviewer #2: Yes

Reviewer #3: No

4. Is the manuscript presented in an intelligible fashion and written in standard English?

Reviewer #1: Yes

Reviewer #2: Yes

Reviewer #3: Yes

5. Review Comments to the Author

Reviewer #1: Dear authors

Some comments and suggestions for further clarification are listed below.

1- Introduction:

a. Add related studies

b. Mention the knowledge gaps of the previous related studies and highlight the importance of your research

c. Write the hypothesis or null hypothesis

d. Clarify why open-ended oral questions are suitable for low-literacy populations

2- Results:

a. Tables are missing from the manuscript

3- Discussion:

a. Highlight the advantages of KROHL-S over other tools (oral administration, literacy independence).

b. Mention limitations (manual scoring, selection bias, generalizability)

4- Conclusion

a. Emphasize practical applications in clinical practice and patient education.

b. Highlight its contribution to addressing oral health disparities

5- Comprehensive English editing is needed

Good Luck with your submission

Reviewer #2: The article complies with PLOS One standards.

The study was well conducted, concise and objective. Also, it is relevant due to the high number of Spanish-speaking people living in the United States. Suggestions were made below in order to contribute to the publication.

Summary

The abstract is well structured and with all the information necessary to understand the study.

Introduction

On lines 84, 85 and 86: “Research consistently demonstrates a strong correlation between limited health literacy and poor oral health outcomes.” Cite the reference for this statement.

On lines 91 to 94: “Health literacy not only affects individual knowledge, attitudes and behaviors but also shapes skills related to disease prevention and self-management. In the clinical setting, health knowledge, a key component of health literacy, serves as an effective measure and it is widely used to assess public health interventions.” Cite the reference used for this statement.

At the end of the introduction, there is not a paragraph on the importance of validating this questionnaire for the Hispanic population. What contributions can this survey on oral health knowledge among Spanish-speaking adults bring to these people living in the United States?

Methodology

It is written objectively and with all the information necessary for the reproduction of the study.

Discussion

On lines 245 and 246: “In contrast to other oral health literacy measures, KROHL-S evaluates condition-specific and domain of knowledge specific aspects of the most common oral conditions.” Cite the reference that supported this statement.

On lines 272 to 274: “To address this issue, we are currently in the process of incorporation different technologies to capture the responses using voice-to-text applications, as well as training different AIs to automate the scoring.” Explain what these technologies are to capture the responses using voice-to-text applications, as well as training different AIs to automate the scoring, in addition to the types of studies that are being developed.

Reviewer #3: Dear Authors,

I have read your manuscript with interest and would like to offer the following comments.

General comment

I note that you have cited only 13 references. Although this journal does not set a minimum number of citations, I consider that the bibliography should be expanded, particularly in the Introduction and Discussion. Additional references to previous instruments, reporting guidelines, and relevant methodological sources would strengthen the rationale and lend greater weight to the claims made in the manuscript.

Introduction

The manuscript conflates two related but non-equivalent concepts, which may create confusion about what exactly the KROHL-S measures and what conclusions can reasonably be drawn. It would therefore be helpful to delineate the construct more clearly: oral health knowledge versus oral health literacy. While KROHL-S is presented as a knowledge tool, much of the framing draws on the oral health literacy literature. A clearer conceptual framework is recommended, including explicit justification for focusing on knowledge (rather than other literacy dimensions), as the current framing—grounded in health literacy and inequities—may lead readers to infer that the instrument measures oral health literacy more broadly.

I also perceive a gap regarding Spanish-language instruments. It would be useful to clarify which prior instruments exist in Spanish (if any), why they do not capture the intended construct or open-ended format, and what KROHL-S adds compared with available alternatives.

Methods

As data collection appears to have been conducted at a single centre, there is potential sampling bias. This should be explicitly acknowledged as a limitation in the Methods section as well.

No formal reliability reporting is provided to document how consensus between raters was established (e.g., ICC/kappa, double-scoring of a subsample). Given the open-ended nature of the responses and the scoring procedure, this information is important.

In addition, no a priori sample size calculation or formal sample size rationale is presented. As the study includes psychometric analyses (including CFA), it would be important to add an explicit “Sample size determination/rationale” section stating whether the final sample (n=177) resulted from convenience sampling and, if so, justifying its adequacy in relation to model complexity (number of factors/items and estimated parameters), the estimator used, and the handling of missing data.

Results

Regarding interpretation of the total score: if the maximum score is 38, a mean of 8.34 suggests a possible floor effect. I would recommend providing the score distribution (e.g., histogram), the proportion of zero/low scores by domain, and item-level analyses (difficulty; floor/ceiling effects).

With respect to the CFA, the manuscript should avoid characterising model fit as “adequate” without qualification. Values of CFI=0.86 and TLI=0.83 are typically below commonly used thresholds, and RMSEA is statistically significant (p=0.044). These results should be reported in greater detail and interpreted consistently with the Discussion, avoiding overstatement where possible.

Please also report effect sizes and ideally 95% confidence intervals, and consider adjustment for relevant covariates (e.g., age, years in the USA, education/literacy, insurance status).

Discussion

The tone of the psychometric conclusions should be moderated, as the manuscript itself notes that “the fit of the overall scale structure was average”. The final summary and the Abstract should therefore avoid presenting the CFA results as “adequate” without nuance.

The manuscript mentions the possibility of voice-to-text and “training different AIs to automate scoring” as future work. It would be helpful to state explicitly:

that this was not evaluated in the present study;

what validation would be required (e.g., transcription error, dialect-related bias, comparison against human raters, privacy considerations); and

that any automation would necessitate a new study of validity and reliability.

Conclusions

Please avoid generalisations beyond the sample (single-centre, convenience sampling). In addition, the conclusions should be aligned with the CFA results (i.e., average/moderate fit).

Data repository

PLOS ONE requires a Data Availability Statement and that the underlying data be available without restriction. The manuscript indicates “available on request”, which is not acceptable under the journal’s policy.

6. PLOS authors have the option to publish the peer review history of their article (what does this mean?). If published, this will include your full peer review and any attached files.

Reviewer #1: No

Reviewer #2: No

Reviewer #3: No

---

## [Author Response · Author response to Decision Letter 1]

23 Feb 2026

We want to thank the reviewers for their conscientious and helpful comments.

We address them all here, and the revised version of the manuscript.

Thank you very much for the insightful review.

Reviewer #1: Dear authors, Some comments and suggestions for further clarification are listed below.

1- Introduction:

a. Add related studies.

We expanded the background information and added all relevant studies. Please refer to citations 8-16

b. Mention the knowledge gaps of the previous related studies and highlight the importance of your research

We addressed this issue in the paragraph starting at 112

c. Write the hypothesis or null hypothesis

Refer to the statement starting at 126

d. Clarify why open-ended oral questions are suitable for low-literacy populations

Collecting the patient’s own words was critical to the development of the KROHL scale, but also the use of open-ended questions is a recommended strategy for communication among individuals with low literacy.

2- Results:

a. Tables are missing from the manuscript.

All the tables are now included.

3- Discussion:

a. Highlight the advantages of KROHL-S over other tools (oral administration, literacy independence).

Please refer to the paragraph starting at 278

b. Mention limitations (manual scoring, selection bias, generalizability)

Limitations are presented in the paragraph starting at 302

4- Conclusion

a. Emphasize practical applications in clinical practice and patient education.

Please refer to the paragraph starting at 325

b. Highlight its contribution to addressing oral health disparities

Please refer to 329

5- Comprehensive English editing is needed

It was addressed

Good Luck with your submission

Reviewer #2:

The article complies with PLOS One standards. The study was well conducted, concise and objective. Also, it is relevant due to the high number of Spanish-speaking people living in the United States. Suggestions were made below in order to contribute to the publication.

Summary: The abstract is well structured and with all the information necessary to understand the study.

Introduction. On lines 84, 85 and 86: “Research consistently demonstrates a strong correlation between limited health literacy and poor oral health outcomes.” Cite the reference for this statement.

The references are now included 2,6

On lines 91 to 94: “Health literacy not only affects individual knowledge, attitudes and behaviors but also shapes skills related to disease prevention and self-management. In the clinical setting, health knowledge, a key component of health literacy, serves as an effective measure and it is widely used to assess public health interventions.” Cite the reference used for this statement.

Please refer to the paragraphs starting at 107 and 112.

At the end of the introduction, there is not a paragraph on the importance of validating this questionnaire for the Hispanic population. What contributions can this survey on oral health knowledge among Spanish-speaking adults bring to these people living in the United States?

Please refer to the paragraph starting at 126. (also 85 and 112)

Methodology It is written objectively and with all the information necessary for the reproduction of the study.

Discussion On lines 245 and 246: “In contrast to other oral health literacy measures, KROHL-S evaluates condition-specific and domain of knowledge specific aspects of the most common oral conditions.” Cite the reference that supported this statement.

Please refer to the paragraph starting at 112 where we discussed other tools and the statements based on citations 11 and 21.

On lines 272 to 274: “To address this issue, we are currently in the process of incorporation different technologies to capture the responses using voice-to-text applications, as well as training different AIs to automate the scoring.” Explain what these technologies are to capture the responses using voice-to-text applications, as well as training different AIs to automate the scoring, in addition to the types of studies that are being developed.

Please refer to the paragraph starting at 314

Reviewer #3: Dear Authors,I have read your manuscript with interest and would like to offer the following comments.

General comment

I note that you have cited only 13 references. Although this journal does not set a minimum number of citations, I consider that the bibliography should be expanded, particularly in the Introduction and Discussion. Additional references to previous instruments, reporting guidelines, and relevant methodological sources would strengthen the rationale and lend greater weight to the claims made in the manuscript.

We are now providing a more in-depth background information leading to the creation of KROHL, including our previous steps starting at 90. The number of relevant citations is now 36.

Introduction

The manuscript conflates two related but non-equivalent concepts, which may create confusion about what exactly the KROHL-S measures and what conclusions can reasonably be drawn. It would therefore be helpful to delineate the construct more clearly: oral health knowledge versus oral health literacy. While KROHL-S is presented as a knowledge tool, much of the framing draws on the oral health literacy literature. A clearer conceptual framework is recommended, including explicit justification for focusing on knowledge (rather than other literacy dimensions), as the current framing—grounded in health literacy and inequities—may lead readers to infer that the instrument measures oral health literacy more broadly.

This point was addressed in the paragraph starting at 107

I also perceive a gap regarding Spanish-language instruments. It would be useful to clarify which prior instruments exist in Spanish (if any), why they do not capture the intended construct or open-ended format, and what KROHL-S adds compared with available alternatives.

Available tools are discussed in the paragraph starting at 98

Methods

As data collection appears to have been conducted at a single centre, there is potential sampling bias. This should be explicitly acknowledged as a limitation in the Methods section as well.

Please refer to the paragraph starting at 305.

No formal reliability reporting is provided to document how consensus between raters was established (e.g., ICC/kappa,double-scoring of a subsample). Given the open-ended nature of the responses and the scoring procedure, this information is important. In addition, no a priori sample size calculation or formal sample size rationale is presented

ICC results are now reported. Please refer to 245

As the study includes psychometric analyses (including CFA), it would be important to add an explicit “Sample size determination/rationale “section stating whether the final sample (n=177) resulted from convenience sampling and, if so, justifying its adequacy in relation to model complexity (number of factors/items and estimated parameters), the estimator used, and the handling of missing data.

This information is now available. Please refer to 214

Results

Regarding interpretation of the total score: if the maximum score is 38, a mean of 8.34 suggests a possible floor effect. I would recommend providing the score distribution (e.g., histogram), the proportion of zero/low scores by domain, and item-level analyses (difficulty; floor/ceiling effects).

The histogram information for each subscale and the overall KROHL is now reported, starting at 229. Since the concept of a right answer is not applicable to this instrument, item difficulty was not calculated.

With respect to the CFA, the manuscript should avoid characterising model fit as “adequate” without qualification. Values of CFI=0.86 and TLI=0.83 are typically below commonly used thresholds, and RMSEA is statistically significant (p=0.044). These results should be reported in greater detail and interpreted consistently with the Discussion, avoiding overstatement where possible.

The language was removed.

Please also report effect sizes and ideally 95% confidence intervals, and consider adjustment for relevant covariates (e.g., age, years in the USA, education/literacy, insurance status).

We have included confidence intervals where appropriate and show all in Table 3.

Discussion:

The tone of the psychometric conclusions should be moderated, as the manuscript itself notes that “the fit of the overall scale structure was average”. The final summary and the Abstract should therefore avoid presenting the CFA results as “adequate” without nuance.

These have been removed.

The manuscript mentions the possibility of voice-to-text and “training different AIs to automate scoring” as future work. It would be helpful to state explicitly that this was not evaluated in the present study; what validation would be required (e.g., transcription error, dialect-related bias, comparison against human raters, privacy considerations); and that any automation would necessitate a new study of validity and reliability.

Please review the paragraph starting at 314

Conclusions

Please avoid generalisations beyond the sample (single-centre, convenience sampling). In addition, the conclusions should be aligned with the CFA results (i.e., average/moderate fit).

Please review the conclusion starting at 328

Data repository PLOS ONE requires a Data Availability Statement and that the underlying data be available without restriction. The manuscript indicates “available on request”, which is not acceptable under the journal’s policy.

The data is now available through the following link: https://doi.org/10.58153/frv59-hzf81

---

## [Decision Letter · Decision Letter 1]

15 Apr 2026

PONE-D-25-30680R1Breaking barriers:validation of a Spanish oral health knowledge tool to enhance patient-provider communicationPLOS One

Dear Dr. Spivakovsky,

Thank you for submitting your manuscript to PLOS ONE. After careful consideration, we feel that it has merit but does not fully meet PLOS ONE’s publication criteria as it currently stands. Therefore, we invite you to submit a revised version of the manuscript that addresses the points raised during the review process.

We look forward to receiving your revised manuscript.

Kind regards,

Yolanda Malele-Kolisa, BDS, MPH, MDent, PhD

Academic Editor

PLOS One

Journal Requirements:

Additional Editor Comments (if provided):

justify and be explicit and specific with the confirmatory factor analysis . A more explicit explanation linking the final sample size to the complexity of the tested model would be necessary to support the robustness of the CFA results.

Second, there is still an inconsistency in the interpretation of model fit. While the Discussion adopts a more cautious tone, the Abstract continues to describe the CFA results as indicating “adequate fit.” Given the reported indices, this wording appears stronger than what the data support. The interpretation of model fit should be aligned throughout the manuscript to ensure internal consistency and appropriate methodological caution.

Reviewers' comments:

Reviewer's Responses to Questions

**Comments to the Author**

1. If the authors have adequately addressed your comments raised in a previous round of review and you feel that this manuscript is now acceptable for publication, you may indicate that here to bypass the “Comments to the Author” section, enter your conflict of interest statement in the “Confidential to Editor” section, and submit your "Accept" recommendation.

Reviewer #1: (No Response)

Reviewer #2: All comments have been addressed

Reviewer #3: (No Response)

2. Is the manuscript technically sound, and do the data support the conclusions?

Reviewer #1: (No Response)

Reviewer #2: Yes

Reviewer #3: Partly

3. Has the statistical analysis been performed appropriately and rigorously? 

Reviewer #1: (No Response)

Reviewer #2: (No Response)

Reviewer #3: No

4. Have the authors made all data underlying the findings in their manuscript fully available?

Reviewer #1: (No Response)

Reviewer #2: (No Response)

Reviewer #3: Yes

5. Is the manuscript presented in an intelligible fashion and written in standard English?

Reviewer #1: (No Response)

Reviewer #2: Yes

Reviewer #3: Yes

6. Review Comments to the Author

Reviewer #1: (No Response)

Reviewer #2: General comment

No formal reliability reporting is provided to document how consensus between raters

was established (e.g., ICC/kappa,double-scoring of a subsample). In this comment: "Internal consistency was good for the general scale and for the oral cancer and malocclusion subscales, and acceptable for dental caries and periodontal disease." It is necessary to include the values Internal consistency".

In addition, it is necessary that the research has not been published in any journal before and complies with the ethical principles of research with human beings.

Reviewer #3: Thank you for your careful revisions. The manuscript has improved substantially; however, two methodological issues remain insufficiently clarified.

First, although a “Sample Size Rationale” section has been added, the justification provided for the adequacy of the sample in relation to the confirmatory factor analysis remains too general. A more explicit explanation linking the final sample size to the complexity of the tested model would be necessary to support the robustness of the CFA results.

Second, there is still an inconsistency in the interpretation of model fit. While the Discussion adopts a more cautious tone, the Abstract continues to describe the CFA results as indicating “adequate fit.” Given the reported indices, this wording appears stronger than what the data support. The interpretation of model fit should be aligned throughout the manuscript to ensure internal consistency and appropriate methodological caution.

Addressing these points would further strengthen the transparency and coherence of the study.

7. PLOS authors have the option to publish the peer review history of their article (what does this mean?). If published, this will include your full peer review and any attached files.

Reviewer #1: No

Reviewer #2: No

Reviewer #3: No

---

## [Author Response · Author response to Decision Letter 2]

21 Apr 2026

1. No formal reliability reporting is provided to document how consensus between raters was established (e.g., ICC/kappa,double-scoring of a subsample).

In our previous revision, we included the ICC to assess interrater reliability (see lines 209, 245-246).

2. In this comment: "Internal consistency was good for the general scale and for the oral cancer and malocclusion subscales, and acceptable for dental caries and periodontal disease." It is necessary to include the values Internal consistency".

These were in table 2, but we forgot to add them directly in-line. We’ve done so in our revision.

3. In addition, it is necessary that the research has not been published in any journal before and complies with the ethical principles of research with human beings.

This information was already provided during submission as it is required by publisher.

4. First, although a “Sample Size Rationale” section has been added, the justification provided for the adequacy of the sample in relation to the confirmatory factor analysis remains too general. A more explicit explanation linking the final sample size to the complexity of the tested model would be necessary to support the robustness of the CFA results.

We’ve added further discussion on these limitations (lines 313-317)

5. Second, there is still an inconsistency in the interpretation of model fit. While the Discussion adopts a more cautious tone, the Abstract continues to describe the CFA results as indicating “adequate fit.” Given the reported indices, this wording appears stronger than what the data support. The interpretation of model fit should be aligned throughout the manuscript to ensure internal consistency and appropriate methodological caution.

Thank you for catching this, this has been revised to be consistent throughout.

---

## [Editor Report · Decision Letter 2]

13 May 2026

Breaking barriers:validation of a Spanish oral health knowledge tool to enhance patient-provider communication

PONE-D-25-30680R2

Dear Dr. Spivakovsky,

We’re pleased to inform you that your manuscript has been judged scientifically suitable for publication and will be formally accepted for publication once it meets all outstanding technical requirements.

Kind regards,

Prof Yolanda Malele-Kolisa, BDS, MPH, MDent, PhD

Academic Editor

PLOS One

Additional Editor Comments (optional):

Line 348

Replace 'Citations' with References
---

## [Editor Report · Acceptance letter]

PONE-D-25-30680R2

PLOS One

Dear Dr. Spivakovsky,

I'm pleased to inform you that your manuscript has been deemed suitable for publication in PLOS One. Congratulations! Your manuscript is now being handed over to our production team.

Kind regards,

on behalf of

Prof Yolanda Malele-Kolisa

Academic Editor

PLOS One